# Study on the Optimal Treatment Condition Control of Photothermal Therapy under Various Cooling Time Ratios of Lasers

**DOI:** 10.3390/ijms232214266

**Published:** 2022-11-17

**Authors:** Donghyuk Kim, Hyunjung Kim

**Affiliations:** Department of Mechanical Engineering, Ajou University, Suwon-si 16499, Gyeonggi-do, Republic of Korea

**Keywords:** photothermal therapy, squamous cell carcinoma, apoptosis, numerical, gold nanoparticles, NIR laser, thermal damage, heat transfer, Monte Carlo method, cooling time ratio

## Abstract

Photothermal therapy is a treatment technique that has attracted attention as an alternative to conventional surgical techniques. It is based on the photothermal effect, wherein light energy is converted into thermal energy, and facilitates rapid recovery after treatment. This study employed various laser irradiation conditions and presented conditions with the optimal treatment effects through a numerical analysis based on heat transfer. A skin layer comprising four stages containing squamous cell carcinoma was targeted, and the treatment effect was confirmed by varying the heating conditions of the laser and volume fraction of gold nanoparticles. The therapeutic effect was confirmed through both the apoptosis retention ratio, which quantitatively estimated the degree of maintenance of the apoptosis temperature range within the tumor, and the thermal hazard retention value, which quantitatively calculates the amount of thermal damage to the surrounding normal tissues. Finally, the optimal treatment conditions were determined based on the laser intensity, cooling time ratio, and volume fraction of injected gold nanoparticles through numerical analysis.

## 1. Introduction

Photothermal therapy is a new anticancer treatment method that uses photothermal effects. The photothermal effect, which is the basic principle of photothermal therapy, is the phenomenon wherein light energy is converted into thermal energy when irradiated on a medium [1,2]. Photothermal therapy is a treatment technique based on cell death due to the temperature rise among various cell death mechanisms through this effect. Photothermal therapy has been reported to aid in faster recovery after treatment, and involves a lower risk of secondary infection because of the absence of bleeding compared with conventional anticancer treatment techniques [3,4,5]. However, in general photothermal therapy, research is focused primarily on skin cancer due to the limited skin penetration depth of the laser, which is represented by the light energy [6,7,8].

Photothermal therapy generally utilizes a laser whose wavelength and intensity can be easily controlled. If laser radiation at visible wavelengths irradiates the skin, skin cancer on the skin surface can be treated; however, the high light absorption coefficient of the biological tissue itself may cause unnecessary thermal damage to the surrounding normal tissue [9,10]. In addition, treatment of tumors occurring inside the skin layer is difficult because most of the light energy is absorbed by the surface of the skin. In contrast, when near-infrared wavelengths are used, light energy can reach tumors generated deep in the skin layer. However, due to the low light absorption coefficient of the biological tissue, the temperature range that facilitates such a treatment cannot be attained [11].

To solve this problem, a light absorption enhancer that increases the light absorption coefficient at a specific wavelength is used for treatment [12,13,14]. The light absorption enhancer increases light absorption at a given wavelength via localized surface plasmon resonance (LSPR), a phenomenon that occurs at a specific wavelength. LSPR is an optical phenomenon that occurs in noble metals of nano sized, wherein an electron cluster of particles smaller than the wavelength of a light source resonates with the wavelength of light to enhance the electromagnetic field, thus resulting in increased light absorption [15]. Light absorption enhancers can be manufactured in various forms, among them, gold nanoparticles (GNPs) are predominantly used because they are harmless to the human body [16,17].

Photothermal therapy induces the death of tumor tissues through a rise in temperature. Death caused by the temperature of the biological tissue is largely classified into apoptosis and necrosis [18,19]. Necrosis occurs at temperatures above 50 °C and presents a risk of cancer metastasis due to the outward leakage of substances inside the tissue. In contrast, apoptosis, which is known to occur between 43 and 50 °C, kills itself without affecting the surrounding tissues. Therefore, the temperature range that facilitates apoptosis must be maintained by controlling the treatment conditions such as laser intensity [20].

Current research on photothermal therapy is being conducted using various methods. Zhang et al. [21] conducted a study on photothermal therapy using nanocomposites (AuNRs@Cur) by combining silica-coated gold nanorods (AuNRs) and curcumin (Cur). They reported that AuNRs@Cur exhibited higher photothermal conversion efficiency compared with single Cur and AuNRs, thereby increasing the utilization rate of photosensitizers and significantly improving the bactericidal effect of bacteria. In addition, the cytotoxicity and hemolytic activity of AuNRs@Cur were negligible. Hai et al. [22] studied the photothermal effect of the epidermal growth factor receptor monoclonal antibody (EGFRmAb) and variant AuNRs to induce tumor cell death in an animal model of laryngeal squamous cell carcinoma. They confirmed that the conjugate of EGFRmAb and AuNRs selectively enters laryngeal squamous cell carcinoma; additionally, they analyzed the treatment results of nude mice by changing the irradiation conditions of the laser used for treatment. Reportedly, treatment with EGFRmAb-AuNRs inhibited the growth of Hep-2 and CNE-2 cells, but not normal epithelial cells. After injection into the tumor, AuNRs irradiated with near-infrared lasers effectively inhibiting tumor growth in nude mice without inducing toxic effects. In addition, the apoptosis and necrosis of tumor cells were maximally expressed under laser intensity of 3 W/cm^2^ and AuNR concentration of 280 μg/kg. Yim et al. [23] utilized polydopamine (PDA, a synthetic melanin with broad absorption and high photothermal conversion ability), to prevent a reduction in photothermal efficiency due to the melting effect that occurs, while general AuNRs lose their LSPR properties upon laser irradiation. In this study, an enhanced photomediated theranostic agonist was developed using AuNR@PDA coated with PDA on AuNRs. In vitro experiments confirmed that 95% of SKOV3 ovarian cancer cells were removed when PDA was coated on very small AuNRs and the treatment effect was more effective compared with that obtained using very small AuNRs (74%) and large AuNRs (66%).

Previous studies have indicated that the study of photothermal therapy using various methods is in progress. Currently, photothermal therapy research is conducted in various ways, and recent research is focused on improving the performance of light absorption enhancers to improve the photothermal conversion efficiency. Accordingly, various types of light absorption enhancers are synthesized and manufactured to confirm the light heat conversion efficiency. However, although the photothermal effect, which is one of the heat transfer phenomena, is a key mechanism for photothermal therapy, theoretical analysis is insufficient. The correlation between apoptosis and the amount of thermal damage to the surrounding normal tissues, which is commonly used in biology, has not yet been reported in study from the perspective of heat transfer. In addition, only the treatment results under specific conditions have been confirmed experimentally. Accordingly, in this study, heat transfer theory through numerical analysis was used to confirm the temperature distribution of squamous cell carcinoma in the skin layer and surrounding normal tissues under laser irradiation. In addition, this study aimed at presenting conditions through numerical analysis that produce the optimal treatment effects by deriving the photothermal therapy effect according to the conditions of various lasers and the volume fraction of injected GNPs through the apoptotic variable proposed by Kim et al. [24] and Kim et al. [25].

## 2. Results and Discussion

### 2.1. Temperature Variation in Biological Tissue

In this study, the temperature distribution inside the biological tissue was confirmed by performing photothermal therapy to analyze how long the temperature range at which apoptosis occurs is maintained through numerical analysis. The situation where squamous cell carcinoma occurred inside the human skin layer was modeled numerically, and the temperature change caused by irradiating the laser to the tumor site was confirmed.

Figure 1 shows the temperature variation over time at specific points in the tumor and normal tissues when the laser intensity (*P_l_*) and volume fraction of the injected GNPs (*f_v_*) were 200 mW and 10^−3^, respectively. The variable $\varphi_{c}$ is the ratio of cooling time to heating time. In this study, after fixing the heating time to 30 s, the cooling time was changed and the temperature change in the medium according to each cooling time ratio was analyzed. Figure 1a shows the temperature variation in the tumor tissue at a depth of 0.1 mm from the central point, and the green area indicates the temperature range where apoptosis is known. In the absence of the cooling time, the temperature continued to rise, and the time to maintain the apoptosis temperature range was only approximately 20–70 s after laser irradiation. In addition, the degree of maintenance of the temperature range of apoptosis varied depending on the cooling time. Figure 1b shows the temperature variation in normal tissue at a depth of 2.5 mm from the center, and the yellow part shows the range where cell apoptosis does not occur. The graph confirmed that the degree of maintenance of the temperature range corresponding to apoptosis changed with $\varphi_{c}$. Based on these points, the effect of photothermal therapy was quantitatively identified by confirming the temperature distribution in the tumor and normal tissues for various *P_l_*, $\varphi_{c}$, and *f_v_*, and the conditions for the optimal therapeutic effect are presented as follows.

### 2.2. Apoptosis Retention Ratio

Apoptosis generally refers to the self-destruction of cells without causing biological effects such as metastasis, destruction, or deformation to surrounding tissues; it is known to occur between 43 and 50 °C from a temperature perspective. Herein, photothermal therapy was performed to maximize apoptosis. The apoptotic variables proposed by Kim et al. [24] and Kim et al. [25] were used to quantitatively determine the extent of apoptosis. Among the three apoptotic variables, the apoptosis retention ratio ($\theta_{A}^{*}$), which quantitatively confirms the degree of maintenance of the apoptosis temperature range during treatment, is the average value of the apoptosis ratio ($\theta_{A}$), which represents the ratio of the total volume of the tumor to the volume corresponding to the apoptosis temperature range, over the total treatment time. By performing treatment using this variable, the degree of maintenance of the apoptosis temperature range in the tumor tissue was quantitatively identified, and the degree of apoptosis in the tumors was indirectly confirmed.

Figure 2 shows $\theta_{A}^{*}$ according to the various $\varphi_{c}$ values for each *f_v_*. The graph shows that as $\varphi_{c}$ increased, *P_l_* (where $\theta_{A}^{*}$ reached the maximum) increased. As shown in Figure 3, as $\varphi_{c}$ increased, the total heating time during the total treatment time decreased; as the cooling time increased, more laser heat was required to heat the tumor tissue to the temperature range where apoptosis occurs. In addition, the results confirmed that as *f_v_* increased, *P_l_* (where $\theta_{A}^{*}$ reached a maximum) decreased. This is because as *f_v_* increases, the tumor tissue absorbs much more heat. Therefore, even if less laser heat is injected, it reached the apoptosis temperature range. Comparative analysis of all cases confirmed that the temperature band of apoptosis in the tumor was maintained at a maximum when $\varphi_{c}$, *f_v_*, and *P_l_* were 0.5, 10^−6^, and 200 mW, respectively.

### 2.3. Thermal Hazard Retention Value

When the skin surface is irradiated by laser, the heat of the laser is transferred to the biological tissue, and light absorption occurs primarily in the tumor tissue due to the increased light absorption coefficient of the tumor tissue injected with GNPs. Subsequently, the absorbed heat is transferred to the surrounding normal tissues through conduction, and if the laser intensity increases excessively, the temperature of the surrounding normal tissues rises to the temperature range corresponding to cell death. To confirm this effect, this study quantitatively determined the amount of thermal damage to the surrounding normal tissues using the thermal hazard retention value ($\theta_{H}^{*}$) [25], which is a variable obtained by averaging the thermal hazard value ($\theta_{H}$) over time. $\theta_{H}$ quantitatively confirms the amount of thermal damage by assigning weights to various phenomena that occur according to the temperature in biological tissue. Thus, the extent of thermal damage in the surrounding normal tissues during treatment can be quantitatively confirmed.

Figure 4 shows $\theta_{H}^{*}$ according to various $\varphi_{c}$ for each *f_v_*. The graph confirms that $\theta_{H}^{*}$ increased with *P_l_*, whereas $\theta_{H}^{*}$ decreased with $\varphi_{c}$. This is because when *P_l_* increased, more heat was absorbed from the tumor tissue and transferred to the surrounding normal tissue. In addition, as $\varphi_{c}$ increased, the heating time of the laser decreased and the cooling time increased; hence, the increase in temperature was less. As *f_v_* increased, the thermal damage was high at the same *P_l_* because the light absorption coefficient of the medium increased with *f_v_*. Because a relatively higher amount of heat was absorbed, the amount of heat transferred to the surrounding normal tissue also increased.

### 2.4. Effective Apoptosis Retention Ratio

Photothermal therapy should maximize apoptosis within the tumor and minimize thermal damage to the surrounding normal tissues. Accordingly, the effect of photothermal therapy was confirmed using the effective apoptosis retention ratio ($\theta_{eff}^{*}$), which simultaneously considers both situations [25]. $\theta_{eff}^{*}$ is calculated as the ratio of $\theta_{A}^{*}$ and $\theta_{H}^{*}$, and it is a variable that quantitatively confirms the degree of apoptosis in the tumor tissue and the amount of thermal damage to the surrounding normal tissue simultaneously. In this study, the conditions for *P_l_*, $\varphi_{c}$, and *f_v_*, which produce the optimal therapeutic effect, were confirmed through the corresponding variables.

Figure 5 shows $\theta_{eff}^{*}$ for various $\varphi_{c}$ according to *f_v_*. The trend of the overall graph was derived in a manner similar to that of $\theta_{A}^{*}$; the results confirmed that *P_l_* and *f_v_*, which produce the optimal treatment effect, exist according to each $\varphi_{c}$. In addition, a comparative analysis of all cases confirmed that the optimal treatment effect was obtained when $\varphi_{c}$, *f_v_*, and *P_l_* were 0.5, 10^−5^, and 160 mW, respectively.

The results indicated that higher laser intensity was required to maintain the apoptosis temperature range of the tumor tissue as the cooling time increased when $\varphi_{c}$ was 0.5 or higher; however, but unnecessary thermal damage occurred due to increased heat transfer to the surrounding normal tissues. This was confirmed by the fact that the optimal point for $\theta_{A}^{*}$, which is a variable confirming the apoptosis temperature band of only the tumor tissue, was derived at a laser intensity of 200 mW, but the optimal point was derived for $\theta_{eff}^{*}$ at 160 mW.

## 3. Materials and Methods

### 3.1. Optical Properties of Gold Nanoparticles

In general, photothermal therapy utilizes GNPs to enhance the low light-absorption rate of biological tissues in the near-infrared region. GNPs increase light absorption at a specific wavelength via the LSPR phenomenon.

The optical properties of GNPs can be calculated using the effective light absorption radius (*r_eff_*) increased by LSPR, the absorption efficiency (*Q*) of the particles, and volume fraction of GNPs in the medium (*f_v_*), as shown in Equation (1) [26]. In this study, AuNRs with an aspect ratio of 6.67 and *r_eff_* of 20 nm were used, and the absorbance efficiency of the AuNRs was calculated using the discrete dipole approximation (DDA) method [27,28].
(1)$$\mu_{a,np}=0.75f_{v}\frac{Q_{a}\;}{r_{eff}},\;\mu_{s,np}=0.75f_{v}\frac{Q_{s}\;}{r_{eff}}$$
(2)$$\mu_{a}=\mu_{a,m}+\mu_{a,np},\;\mu_{s}=\mu_{s,m}+\mu_{s,np}$$

When the absorption and scattering coefficients of GNPs were calculated, the optical properties of the medium containing GNPs were calculated as the sum of the optical properties of the medium and GNPs (Equation (2)) [29].

### 3.2. Modeling for Light Absorption and Scattering in Biological Tissue

When a laser irradiates a biological tissue, its energy is absorbed and scattering occurs simultaneously inside the medium. In the field of bioheat transfer, these behaviors have been analyzed using the Monte Carlo method [30], which performs probabilistic calculations of the absorption and scattering behavior of laser particles in a medium using random numbers.

Monte Carlo method determines the behavior of laser particles in a medium by calculating the distance and angle of movement over time. First, the distance moved per time can be calculated using the total attenuation coefficient ($\mu_{tot}$) of the medium and a randomly determined random number between 0 and 1 (*ξ*), as shown in Equation (3).
(3)$$S=\frac{-\ln(\xi )}{\mu_{t}{}_{ot}}\;(\mu_{tot}=\mu_{a}+\mu_{s})$$

For the moving angles, the azimuth and deflection angles are calculated separately. The azimuth (*ψ*) can be calculated as in Equation (4) using the anisotropy factor (*g*), which determines the direction in which the particles are scattered. The deflection angle *θ* is calculated in two ways, as shown in Equation (5), according to the anisotropy factor.
(4)ψ=2πξ
(5)$$\cos\theta =\{\begin{array}{cc} \frac{1}{2g}\left\{1+g^{2}-{\left[\frac{1-g^{2}}{1-g+2g\xi }\right]}^{2}\right\} & if\;g>0\\ 2\xi -1 & if\;g=0 \end{array}$$

After estimating the azimuth and deflection angles, the direction vectors $\mu_{x}^{\prime }$, $\mu_{y}^{\prime }$, and $\mu_{z}^{\prime }$ at which the photons move in the Cartesian coordinate system can be calculated. If the moving angle is in the vertical direction of the tissue surface, it can be calculated using Equations (6)–(8) according to each axis. Here, $\mu_{x}$,$\;\mu_{y}$, and $\;\mu_{z}$ are the directional cosines for each axis.
(6)$$\mu_{x}^{\prime }=\sin\theta \cos\psi \;(0\;\text{<}\;\theta \;\text{<}\;\pi ,\;0\;\text{<}\;\psi \;\text{<}\;2\pi )$$
(7)$$\mu_{y}^{\prime }=\sin\theta \sin\psi \;(0\;\text{<}\;\theta \;\text{<}\;\pi ,\;0\;\text{<}\;\psi \;\text{<}\;2\pi )$$
(8)$$\mu_{z}^{\prime }=\{\begin{array}{cc} \cos\theta & if\;\mu_{z}>0\\ -\cos\theta & f\;\mu_{z}<0 \end{array}\;(0\;\text{<}\;\theta \;\text{<}\;\pi )$$

If the moving angle is in another direction, it can be calculated using Equations (9)–(11), as follows.
(9)$$\mu_{x}^{\prime }=\frac{\sin\theta }{\sqrt{1-\mu_{z}^{2}}}(\mu_{x}\mu_{z}\cos\psi -\mu_{y}\sin\psi )+\mu_{x}\cos\theta$$
(10)$$\mu_{y}^{\prime }=\frac{\sin\theta }{\sqrt{1-\mu_{z}^{2}}}(\mu_{y}\mu_{z}\cos\psi -\mu_{x}\sin\psi )+\mu_{z}\cos\theta$$
(11)$$\mu_{z}^{\prime }=-\sin\theta \cos\psi \sqrt{1-\mu_{z}^{2}}+\mu_{z}\cos\theta$$

Finally, when the distance and angle of movement are calculated per time, the ratio of the energy decrease per time can be calculated, as shown in Equation (12), where *W* represents the energy weight of the photon and the movement is repeated until *W* converges to zero. The final distribution of the light absorption in the medium can be calculated by repeating this process as many times as the number of photons.
(12)$$\Delta W=W\frac{\mu_{a}}{\mu_{t}{}_{ot}}$$

If the light absorption distribution of the medium is calculated using the Monte Carlo method, the temperature distribution in the medium over time can be calculated using the heat diffusion equation in Equation (13), where *k*, *ρ*, $c_{v}$, and *q* represent the thermal conductivity coefficient, density, specific heat, and amount of heat absorbed by the medium, respectively. The temperature distribution in the medium is calculated using the explicit finite element method, as shown in Equation (14), where *dx, dy*, and *dz* are the length elements of each axis; $P_{l}$ is the intensity of the laser; and *F* is the fluence rate [31].
(13)$$\frac{\partial T}{\partial \tau }=\frac{q+\nabla \cdot (k\nabla T)}{\rho c_{v}}$$
(14)$$\begin{array}{l} \Delta T=\frac{\Delta \tau }{\rho c_{v}}(\mu_{a}FP_{l}dxdydz+(T_{x^{-}}-T)\frac{2kk_{x^{-}}}{k+k_{x^{-}}}\frac{dydz}{dx}+(T_{x^{+}}-T)\frac{2kk_{x^{+}}}{k+k_{x^{+}}}\frac{dydz}{dx}\\ \;\;\;\;+(T_{y^{-}}-T)\frac{2kk_{y^{-}}}{k+k_{y^{-}}}\frac{dxdz}{dy}+(T_{y^{+}}-T)\frac{2kk_{y^{+}}}{k+k_{y^{+}}}\frac{dxdz}{dy}\\ \;\;\;\;+(T_{z^{-}}-T)\frac{2kk_{z^{-}}}{k+k_{z^{-}}}\frac{dxdy}{dz}+(T_{z+}-T)\frac{2kk_{z+}}{k+k_{z^{+}}}\frac{dxdy}{dz}) \end{array}$$

### 3.3. Numerical Conditions

In this study, the situation in which squamous cell carcinoma occurs inside the skin layer consisting of four stages was numerically modeled. The skin consists of the epidermis, reticular dermis, papillary dermis, and subcutaneous fat; the total radius and depth were set to 15 mm and 20 mm, respectively, as shown in Figure 6. In the case of squamous cell carcinoma, the radius and depth were set to 5 and 2 mm, respectively, and it was assumed to have occurred 0.1 mm from the skin surface. A 1064 nm single-wavelength laser with a top-hat distribution and radius of 5 mm was used as the heat source. The laser was assumed to be irradiated in the vertical direction of the tumor tissue. The thickness and thermal and optical properties of the four skin layers and tumor tissues are listed in Table 1. Numerical analysis modeling was verified through the biomimetic phantom experiment reported in our previous study, and an average of 0.1677 was derived for four points by comparing the root mean square error of the experiment and the numerical analysis. This was judged to adequately simulate the actual situation [25].

In this study, numerical analysis was performed by varying the laser intensity and volume fraction of the injected GNPs to derive the optimal conditions for photothermal therapy via laser irradiation of the skin surface. As seen in the study of Jain et al. [39], the absorption and scattering efficiency of gold nanoparticles varied according to various shapes and sizes and the wavelength of the irradiated laser. In this study, to analyze this phenomenon, the optical efficiency of the particles was calculated using the discrete DDA method. Rod-type gold nanoparticles with an effective radius of 20 nm and aspect ratio of 6.67 were used, and various laser cooling times were included among the analysis parameters to generate a thermal confinement effect [40], which refers to the effect of heat conduction in a medium that is not widely distributed, so that heat transfer occurs in a limited range. Thus, if an appropriate cooling time is provided when irradiating the laser, thermal confinement occurs, and heat transfer occurs only in the tumor tissue area, thereby minimizing thermal damage to the surrounding normal tissue. Accordingly, a variable called the cooling time ratio ($\varphi_{c}$), which represents the ratio of the cooling time to the heating time, was used. Note that $\varphi_{c}$ = 0 indicates the continuous application of heat during the treatment time without the cooling time. When $\varphi_{c}$ = 1, heating was applied for the set heating time, and cooling for the same duration of time was repeated during the treatment time. To confirm the treatment effect according to the ratio of the cooling time to the same treatment time, the treatment time and heating time were fixed at 900 s and 30 s, respectively, and all analysis variables are summarized in Table 2.

As described in Section 3.1, the optical properties of the entire medium varied depending on the volume fraction of GNPs. This indicates that the ability of the medium to absorb heat varies, and this study confirmed the therapeutic effect of photothermal therapy according to the volume fraction of various GNPs. The laser used in this study had a wavelength of 1064 nm, and after calculating the optical efficiency of gold nanoparticles at a wavelength of 1064 nm, this was reflected in the optical properties of the tumor tissue. Table 3 summarizes the absorption and scattering coefficients of the tumor tissues according to the volume fraction of injected GNPs.

## 4. Conclusions

In this study, photothermal therapy based on heat transfer was numerically analyzed. Actual human skin layer containing squamous cell carcinoma was implemented through numerical analysis modeling, and the distribution of the heat of the laser transmitted in the medium was confirmed using the Monte Carlo method to confirm the temperature distribution in the medium. Finally, the treatment conditions of photothermal therapy were optimized using an effective apoptosis retention ratio that considered both the apoptosis retention ratio, which quantitatively estimates the extent to which the temperature range at which apoptosis occurs within the tumor is maintained, and the thermal hazard retention value, which determines the amount of thermal damage occurring simultaneously in the surrounding normal tissues.

A numerical analysis was performed by varying the laser intensity and volume fraction of the injected GNPs. In addition, the cooling time ratio of the irradiated laser was added to the variables to generate a thermal confinement effect. Finally, the results confirmed that the optimal treatment effect was achieved when the laser intensity, volume fraction of injected GNPs, and laser cooling time ratio were 160 mW, 10^−5^, and 0.5, respectively. However, even if the tumor tissue attains the apoptosis temperature range, research on the duration for which it must be maintained to induce apoptosis is limited. More accurate photothermal therapy can be performed if the temporal effect for the occurrence of apoptosis is investigated in the future. In addition, since this study was conducted through numerical analysis, it is thought that detailed verification through experiments is necessary.

## Figures and Tables

**Figure 1 ijms-23-14266-f001:**
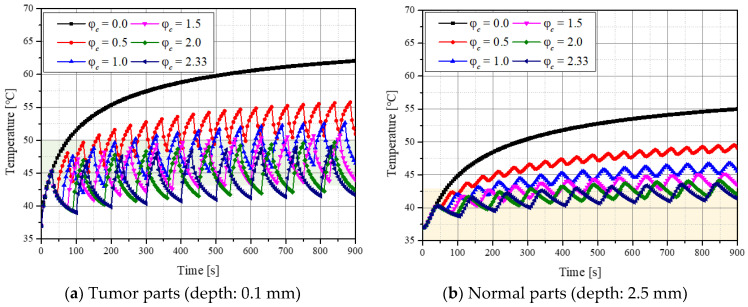
Temperature variation in the tumor and normal tissue over time for various cooling time ratios ($\varphi_{c}$) (*f_v_* = 10^−3^, *P_l_* = 200 mW).

**Figure 2 ijms-23-14266-f002:**
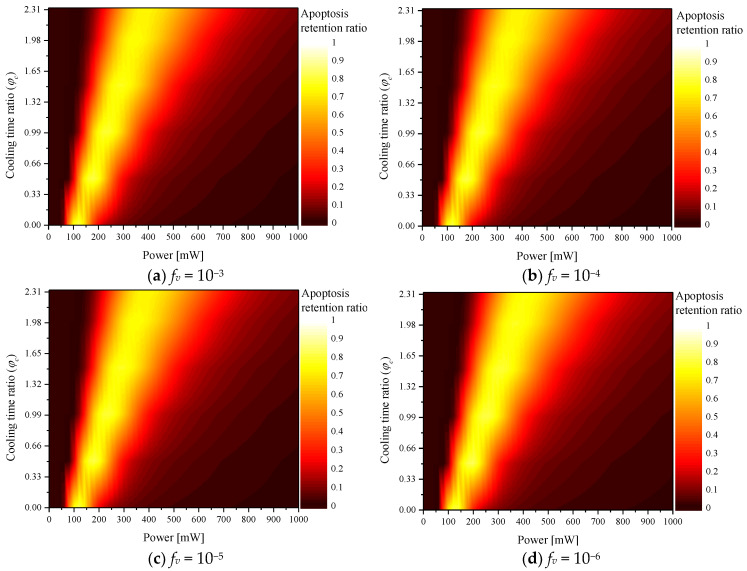
Apoptosis retention ratio ($\theta_{A}^{*}$) for various cooling time ratios ($\varphi_{c}$).

**Figure 3 ijms-23-14266-f003:**
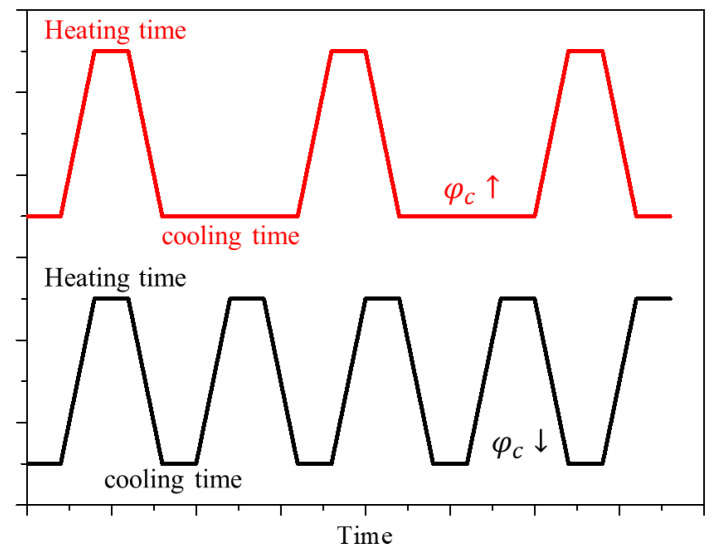
Cooling rate during the treatment time according to the cooling time ratio ($\varphi_{c}$) change.

**Figure 4 ijms-23-14266-f004:**
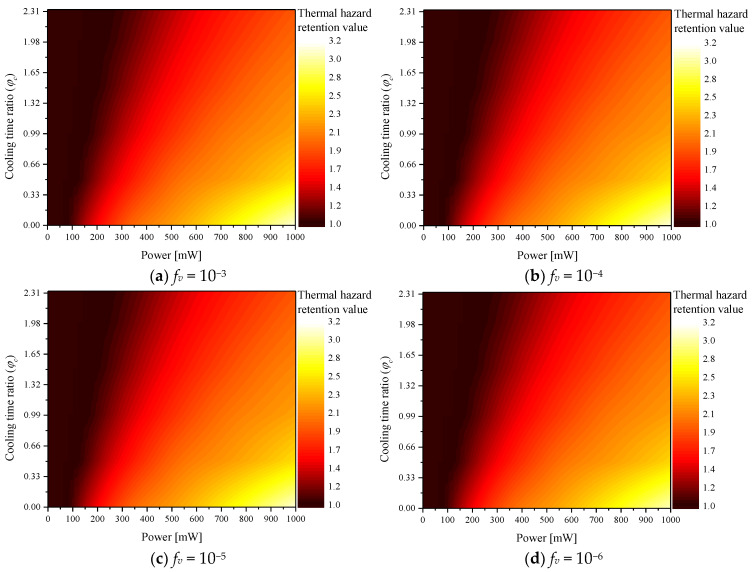
Thermal hazard retention value ($\theta_{H}^{*}$) for various cooling time ratios ($\varphi_{c}$).

**Figure 5 ijms-23-14266-f005:**
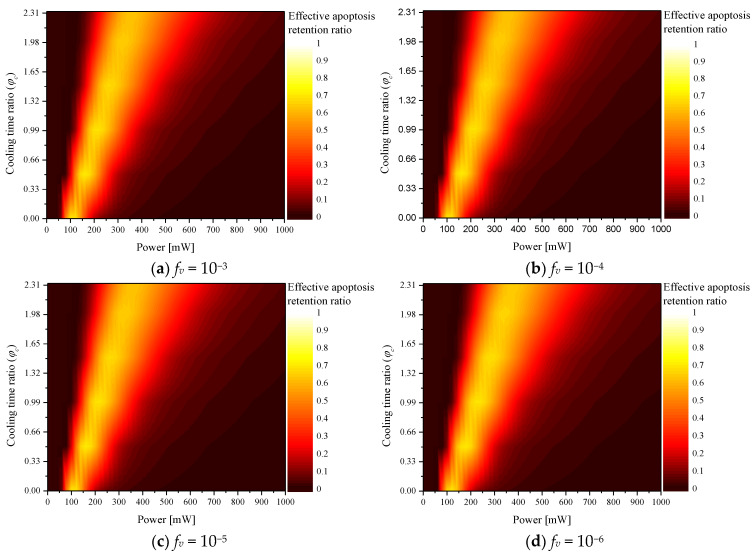
Effective apoptosis retention ratio ($\theta_{eff}^{*}$) for various cooling time ratios ($\varphi_{c}$).

**Figure 6 ijms-23-14266-f006:**
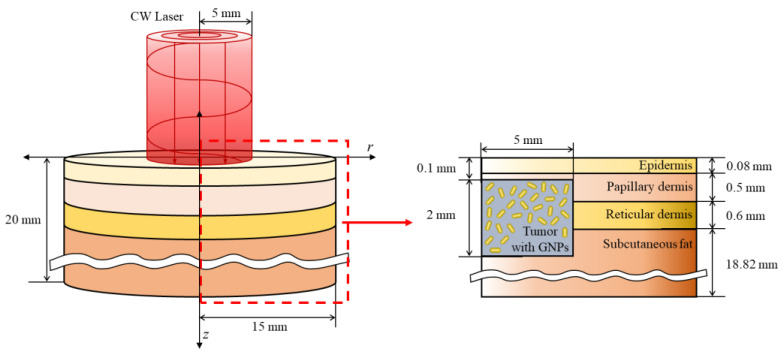
Schematic of the numerical model.

**Table 1 ijms-23-14266-t001:** Depth, thermal, and optical properties of skin layers and tumor [10,32,33,34,35,36,37,38].

	*t* (mm)	*c_v_* (J/kgK)	*ρ* (kg/m^3^)	*k* (W/mK)	*g*	$\mu_{a}$ (1/mm)	$\mu_{s}$ (1/mm)
Epidermis	0.08	3589	1200	0.235	0.8	0.4	45
Papillary dermis	0.5	3300	1200	0.445	0.9	0.38	30
Reticular dermis	0.6	3300	1200	0.445	0.8	0.48	25
Subcutaneous fat	18.82	2500	1000	0.19	0.75	0.43	5
Tumor	2	3421	1070	0.495	0.8	0.047	0.883

**Table 2 ijms-23-14266-t002:** Conditions of the numerical analysis.

Numerical Parameter	Case	Number	Remarks
Treatment time (*τ_tot_*)	900 s	N/A	
Heating time (*τ_h_*)	30 s	N/A	
Laser power (*P_l_*)	0–1000 mW	51	Intv: 20 mW
Volume fraction of GNPs (*f_v_*)	10^−3^–10^−6^	4	Intv: 10^−1^
Cooling time ratio ($\varphi_{c}$)	0, 0.5, 1, 1.5, 2, 2.33	6	

**Table 3 ijms-23-14266-t003:** Optical properties of the tumor with GNPs for the changing volume fraction.

Volume Fraction of GNPs	10^−3^	10^−4^	10^−5^	10^−6^
Absorption coefficient ($\mu_{a}$) (mm^−1^)	557.41	55.78	5.62	0.60
Scattering coefficient ($\mu_{s}$) (mm^−1^)	118.58	12.65	2.06	1.00

## Data Availability

Data sharing is not applicable to this article.

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
