# Peer review of "Study on the Optimal Treatment Condition Control of Photothermal Therapy under Various Cooling Time Ratios of Lasers"

_ijms, 2022, doi:10.3390/ijms232214266_

Round 1
Reviewer 1 Report
The present research aimed at presenting conditions that produce optimal treatment effects by deriving the photothermal treatment effect according to the conditions of various lasers and volume fraction of injected GNPs through the apoptotic variable proposed by Kim et al.[24] and Kim et al.[25].
The authors implemented human skin layer containing cell carcinoma by help of numerical analysis modelling.
In this study the photothermal therapy based on heat transfer was numerically analyzed, by varying the laser intensity and volume fraction of the injected GNPs. The heat of the laser transmitted in the medium was evaluated using the Monte Carlo method. In addition, the cooling time ratio of the irradiated laser was added to the variables to generate a thermal confinement effect.
However, even if the tumor tissue attains the apoptosis temperature range, research on the duration for which it must be maintained to induce apoptosis is limited.
The authors conclude that more accurate photothermal therapy can be performed and the duration effect for the occurrence of apoptosis should be investigated in the future.
Author Response
Thank you for your review.
Reviewer 2 Report
This paper by Donghyuk Kim and Hyunjung Kim is a continued work on numerical analysis of photothermal therapy. The numerical study based on the Monte Carlo modeling unambiguously analyzed the heat transfer process. The result showed that the optimal treatment effect was obtained when laser intensity, cooling time ratio, and volume fraction of injected gold nanoparticles were 160 mW, 0.5, and 10-5 respectively. This research is interesting and has potential application in photothermal therapy. Thus I would recommend publishing it as it is.
Author Response
Thank you for your review.
Reviewer 3 Report
The abstract begins with an incorrect statement: PDT is not based on a thermal effect but on the production of reactive oxygen species upon irradiation of sites at which photosensitizers have accumulated. Moreover, PDT is often used to remove residual tumor remaining at surgical sites. Formation of reactive oxygen species in targeted organelles can lead to apoptosis. Photothermal effects can occur but require a much higher photon flux. An examination of the potential for photothermal effects could, however, be useful but one cannot neglect the well-established pathways that lead to apoptosis via the release of cytochrome c from damaged mitochondria into the cytoplasm.
The authors provide minimal information on methodology. There is no information on the absorbance spectrum of the photosensitizing agent or on its ability to show selective affinity for sites of neoplasia. There is minimal information on irradiation (wavelength and light dose) or how apoptosis was assessed. The term ‘apoptosis retention ratio’ is essentially meaningless. Experimental details are minimal. There is no discussion of how tissue samples were obtained and treated (Table 1). The legend to Fig. 6 mentions 1064 nm irradiation but apparently the effects are solely the result of heating. There is no mention of any photosensitizing agent being present or details concerning the tissue preparation used. This appears to be a study on the effects of hyperthermia and cites a collection of references. From which were these data abstracted?
What is missing: [1] the precise structure of the gold nanoparticles and their absorbance spectrum, [2] evidence (if any) for selective retention of this agent in sites of neoplasia, [3] evidence that the photothermal effect is solely responsible for the anti-tumor effect, [4] is it assumed that production of reactive oxygen species is not occurring or that this will have minimal efficacy compared with photothermal effects?
The figure legends are often uninformative. The meaning of the assorted curves in Fig. 1 is not indicated in the legend. What was the model used? If the heating effect is not confined to sites of neoplasia, is the inference is that tumors are more heat-sensitive than adjoining normal tissue.
Author Response
Thanks for the comment. More detailed information is provided in the attached file. please check.

Round 2
Reviewer 3 Report
This report describes use of gold nanoparticles (GNPs) for promoting thermal effects designed to aid in tumor eradication. Mention is made of GNPs coated with silica and curcumin (AuNRs@Cur), and with EGFR monoclonal antibodies. The term AuNRs is also used. This tends to be confusing since it is not clear exactly which preparation was used in data reported here. The Materials and Methods section discusses optical properties, laser interactions with tissues, heat transfer, skin properties, but there is no information on exactly what was done.
Each figure is supposed to be internally consistent with a legend that describes what is indicated. Fig. 1 does not described what system is used: what animal model, what tumor and what treatment? How is apoptosis identified? The definition of apoptosis in section 2.2 is totally inadequate. Apoptosis has a definite meaning in the context of biochemistry, referring to the activation of caspases to cleave genomic DNA. What does ‘maintain apoptosis’ mean? This is an irreversible process so once initiated, it cannot be reversed. Apoptosis does not need to be ‘maintained’. It is not reversible. How was apoptosis verified in the samples, whatever they might be?
Summary: There is minimal information about what was actually done in the experiments reported.
Author Response
I have confirmed your comment. Thanks for the good comments. Please check the attachment file.
